# The Mitochondrial Calcium Uniporter (MCU): Molecular Identity and Role in Human Diseases

**DOI:** 10.3390/biom13091304

**Published:** 2023-08-25

**Authors:** Donato D’Angelo, Rosario Rizzuto

**Affiliations:** 1Department of Biomedical Sciences, University of Padua, 35131 Padua, Italy; donato.dangelo.1@phd.unipd.it; 2National Center on Gene Therapy and RNA-Based Drugs, 35131 Padua, Italy

**Keywords:** MCU, mitochondrial Ca^2+^ signaling, cancer, cardiovascular diseases, metabolic diseases, skeletal muscle diseases, neurodegenerative disorders

## Abstract

Calcium (Ca^2+^) ions act as a second messenger, regulating several cell functions. Mitochondria are critical organelles for the regulation of intracellular Ca^2+^. Mitochondrial calcium (mtCa^2+^) uptake is ensured by the presence in the inner mitochondrial membrane (IMM) of the mitochondrial calcium uniporter (MCU) complex, a macromolecular structure composed of pore-forming and regulatory subunits. MtCa^2+^ uptake plays a crucial role in the regulation of oxidative metabolism and cell death. A lot of evidence demonstrates that the dysregulation of mtCa^2+^ homeostasis can have serious pathological outcomes. In this review, we briefly discuss the molecular structure and the function of the MCU complex and then we focus our attention on human diseases in which a dysfunction in mtCa^2+^ has been shown.

## 1. Introduction

Ca^2+^ ions are crucial second messengers involved in the regulation of numerous biological processes such as aerobic metabolism, gene transcription, cell proliferation, and physiological mechanisms such as muscle-cell contraction and synaptic plasticity [1,2]. The major Ca^2+^ store in the cell is the endoplasmic reticulum (ER) (sarcoplasmic reticulum, SR, in striated muscle cells). Upon physiological stimuli, it can rapidly release Ca^2+^ in the cytosol through store-resident channels. In this context, mitochondria play a key role in the regulation of cellular Ca^2+^ homeostasis [3]. Indeed, these organelles are able to take up Ca^2+^ after a release from ER/SR or entry from extracellular space. The increase in mitochondrial matrix Ca^2+^ concentration ([Ca^2+^]_mt_) stimulates the activity of three key dehydrogenases of the TCA cycle, thus promoting oxidative phosphorylation (OXPHOS) and, in turn, ATP production [4]. However, an excessive accumulation of mtCa^2+^, a condition known as mtCa^2+^ overload, can be detrimental for the cell. Indeed, it can trigger the opening of the mitochondrial permeability transition pore (mPTP), promoting the release of pro-apoptotic factors and ultimately leading to apoptotic cell death [5]. The molecular machinery that allows the entry of Ca^2+^ inside the mitochondrial matrix is the mitochondrial calcium uniporter complex (MCUc). The discovery of the pore-forming subunit of the MCUc, MCU itself, in 2011 was a major turning point in the field and paved the way for the characterization of the different subunits that compose this sophisticated complex [6,7]. In this review, we will briefly describe the MCU complex structure and the function of the different complex components. In recent years, a complex scenario has emerged due to the different protein expressions of the MCUc components, accompanied by changes in the kinetics of mtCa^2+^ transport between different tissues and cell types. This aspect is also reflected in the relationship of the MCUc to various human diseases. In this work, we wish to provide an overview of different diseases in which the MCUc is involved.

## 2. MCU Complex Structure and Function

The MCU complex is composed of pore-forming subunits, i.e., MCU and its dominant–negative isoform, MCUb, the essential MCU regulator (EMRE), which allows interaction with the MICU mitochondrial calcium uptake regulatory subunits (MICU1, MICU2 and MICU3), and possibly other modulators, such as the mitochondrial calcium uniporter regulator 1 (MCUR1) (Figure 1).

The molecular identity of MCU was discovered in 2011 by two different studies. MCU is a 40-KDa protein that is highly conserved, ubiquitously expressed, and located in the IMM. These reports show that the downregulation of MCU strongly reduces mtCa^2+^ uptake in cells after Ca^2+^ release from the ER after treatment with an IP3-generating agonist. Notably, these changes occur without impinging on mitochondrial morphology and membrane potential. Coherently, MCU overexpression strongly enhances mtCa^2+^ uptake after agonist-induced stimulation [6,7]. Structure analyses revealed that both the N- and C-termini of the protein face the mitochondrial matrix, and that MCU contains two transmembrane domains linked by a highly conserved loop located in the intermembrane space, containing a “DIME” motif with negatively charged amino acids critical for Ca^2+^ permeation [8].

Raffaello et al., in 2013, described an alternative isoform of MCU, MCUb, which is a 33-KDa protein that shares high structural similarity with MCU. However, critical amino acid substitutions in the DIME motif explain why it exerts a dominant–negative effect, reducing the [Ca^2+^]_mt_ rise evoked by the agonist stimulation. The opposite effect on the mtCa^2+^ uptake was obtained by MCUb silencing, confirming its role as a negative regulator of the channel [9]. The MCU/MCUb ratio varies greatly between different mammalian tissues, and this impinges on the intrinsic capacity of mitochondria of a tissue to rapidly accumulate Ca^2+^. Not surprisingly, the heart undergoes repetitive Ca^2+^ spiking, at the risk of mitochondrial Ca^2+^ overload, displays a low MCU/MCUb ratio, while skeletal muscle ensures maximal and sustained metabolic upregulation in phasic responses through a very low expression of MCUb. In general, the great variability in expression and stoichiometry of MCU and MCUb accounts, at least in part, for the wide differences in the MCU currents measured in different mammalian tissues [10]. While these two tissues appear to represent the opposite extremes of the modulation of mitochondrial Ca^2+^ sensitivity by the MCU/MCUb ratio, the expression data of tissues at the mRNA and protein level, and in different physiological and pathological conditions, gradually provide a much broader and finer picture of the wide molecular flexibility of the MCU complex (MCU/MCUb and, as detailed later in this review, the expression of different MICU isoforms) [10,11].

The Essential MCU Regulator (EMRE) is a 10-KDa metazoan-specific protein, identified by quantitative mass spectrometry of the affinity-purified MCU complex components [12]. It is inserted in the IMM by a single transmembrane domain and is required for the interaction between MCU and MICU1, acting as a bridge between the pore-forming and regulatory subunits of the complex. Experiments performed on EMRE knockout cells clearly demonstrate that mtCa^2+^ uptake is impaired, phenocopying the effect of MCU silencing. This evidence enlightens the critical role of EMRE as an essential component of the MCU complex, which is required for its activity [12]. Coherently, the reconstitution of the human MCU protein alone in yeast cells is not sufficient for uniporter activity because the MCU channel is active only if MCU and EMRE are co-expressed [13]. It is not surprising that the proteolytic regulation of EMRE is a finely tuned process essential for the formation of a functional MCU complex [14].

Cryo-EM structure of fungal and metazoan MCU channels revealed that the pore subunit assembles as homo-tetramer [8]. A structure of human MCU complexed with EMRE showed a 4 EMRE:4 MCU stoichiometry, but the stoichiometry in vivo remained unknown [15]. Recently, Payne et al., demonstrated that MCU does not require four EMRE in vivo, with the majority of channels containing two EMRE, but also complexes with one EMRE can be functional [16].

MICU1 was the first member of the MCU complex to be identified in 2010. It is a 54-KDa protein located in the inner mitochondrial membrane space (IMS) where it regulates the activity of the MCU channel in the Ca^2+^-dependent way [17]. Indeed, it contains two EF-hand Ca^2+^-binding domains at its N-terminal sequence. This regulation ensured by MICU1 explains the sigmoidal response to cytCa^2+^ levels of mtCa^2+^ uptake. At low cytCa^2+^ concentrations, MICU1 keeps the channels closed and mtCa^2+^ uptake is negligible, thus avoiding a constant entry of Ca^2+^ into the mitochondrial matrix. However, when cytCa^2+^ concentration increases, MICU1 acts as a cooperative activator of MCU, leading to an exponential increase of the mtCa^2+^ uptake [18]. An alternative splicing isoform of MICU1, MICU1.1, which is characterized by the addiction of a short exon (four amino acids), was shown to be expressed predominantly in skeletal muscle and at a lower level in the brain. Similar to MICU1, MICU1.1 acts as a positive modulator of the channel, leading to even higher increases of the mtCa^2+^ uptake after stimulation. In particular, MICU1.1 binds Ca^2+^ more efficiently than MICU1. When it is heterodimerized with MICU2, it decreases the Ca^2+^-activation threshold of MCU [19].

Two paralogs of MICU1 residing in the IMS contribute to the regulation of mtCa^2+^-handling: MICU2 and MICU3 [20]. MICU2 has an expression pattern similar to MICU1 as it contains two EF-hand domains and interacts with MICU1 and MCU, forming obligate heterodimers with MICU1 [21]. Interestingly, MICU2 protein stability seems to be dependent on the presence of MICU1. Indeed, MICU1 silencing also leads to the loss of MICU2 [22]. Patron et al. proposed a model in which, in resting conditions, MICU1–MICU2 heterodimers act as gatekeepers of the channel due to the predominant inhibitory effect of MICU2. The increase in Ca^2+^ concentration promotes a conformational change leading to the release of MICU2 inhibition and the MICU1-mediated enhanced Ca^2+^ uptake [21]. Conversely, Kamer et al. proposed that both MICU1 and MICU2 act as gatekeepers of the channel, responding to increases in Ca^2+^ concentration with a single cooperative transition [22]. Notably, another study suggests a unifying model in which the cooperative activation of the channel is maintained, but MICU1 and MICU2 have distinct roles: MICU1 alone can mediate gatekeeping and activation of the channel, while MICU2 regulates the threshold of MICU1-mediated inhibition and activation of the channel. This, in the context of MICU1–MICU2 heterodimers, leads to a reduced sensitivity to cytosolic Ca^2+^ concentrations [23].

MICU3 is another MICU1 paralog mainly expressed in the nervous system and, to a lower extent, in skeletal muscle. Similar to MICU1 and MICU2, MICU3 also contains two EF-hand domains. Patron et al. demonstrated that MICU3 forms heterodimers exclusively with MICU1, not with MICU2. Notably, MICU3 has a reduced gatekeeping activity compared to MICU1 as it mediates more rapid responses to elevated cytCa^2+^ levels. Thus, MICU3 is proposed to be a positive modulator of the MCU channel, ensuring mtCa^2+^ uptake in response to fast cytCa^2+^ rises, which is typical of neuronal stimulation [24].

MCUR1 is a 35KDa protein located in the IMM that interacts with MCU [25]. Its downregulation leads to a decrease in mtCa^2+^ uptake and ATP production [26]. In contrast to this view, Paupe et al. showed that MCUR1 is not a regulator of the MCU complex, but it is a cytochrome c oxidase assembly factor [27]. Thus, the precise role of MCUR1 related to mtCa^2+^ homeostasis still remains to be fully elucidated.

## 3. Cardiovascular Diseases

The dysregulation of mitochondrial calcium (mtCa^2+^) homeostasis occurs in acute myocardial ischaemia reperfusion (I/R) injury points to play a critical role of MCU in cardiovascular diseases. Upon reperfusion, mtCa^2+^ overload leads to excessive reactive oxygen species (ROS) production, promoting the opening of the mitochondrial permeability transition pore (PTP), which triggers cardiomyocyte death [28]. Thus, MCU was conceptually expected to be involved in ischemia-dependent sensitization of cardiac muscle to reperfusion damage.

The analysis of hearts from MCU knockout mice showed not only unaffected basal cardiac parameters, but, surprisingly, they were not protected from ischemia/reperfusion damage, as expected of a role of mitochondrial Ca^2+^ loading upstream of the permeability transition pore (PTP) opening. In addition, the hearts of MCU^-/-^ mice were not protected by treatment with the pore desensitizer cyclosporin A (CsA) [29]. The same lack of protection was also observed in a transgenic mouse model in which MCU downregulation was achieved by the overexpression of a dominant-negative form of MCU (DN–MCU) [30]. Overall, these results suggest that the constitutive ablation of MCU (and, hence, the loss of mtCa^2+^ signals) from the embryonic phase could lead to mitochondrial compensatory mechanisms (e.g., in sensitivity to Ca^2+^ or modulators of PTP or prevalence of other cell-death pathways).

To overcome this problem, a mouse model with an acute deletion of MCU in adult cardiomyocytes was generated [31,32]. In contrast to constitutive MCU deletion, the conditional knockout model showed protection from I/R injury and cell death, thus confirming the view of MCU-dependent dysregulated Ca^2+^ signals upstream of PTP opening and myocardial cell death.

Interestingly, MCUb and the mitochondrial Na^+^/Ca^2+^ exchanger (mNCLX) gene expression increased after ischemic damage to the heart, and their overexpression led to a decrease in mtCa^2+^ by limiting the uptake or enhancing the efflux, respectively. This is protective against I/R injury [33,34].

In this context, the role of MICU1 is still controversial. Indeed, while it is protective in the early stages after reperfusion since its knockdown worsens the I/R damage, MICU1 is also observed to be upregulated during the late stages after reperfusion. However, the mechanism behind this increase is still not clear [33,35].

Together, these findings highlight the relevance of MCU modulation as a potential therapeutic approach in the treatment of cardiovascular diseases. However, further studies are needed to translate these findings in a clinical approach.

## 4. Metabolic Diseases

Normal blood glucose concentrations are ensured by glucose-induced insulin secretion from pancreatic β-cells [36]. In this process, the mitochondrial oxidative metabolism plays a key role. Indeed, an increase in the cytosolic ATP level [37] results in the closure of ATP-sensitive K^+^ channels (K_ATP_) [38], leading to plasma-membrane depolarization. This, in turn, promotes the opening of voltage-gated Ca^2+^ channels, allowing Ca^2+^ entry into the cell, ultimately leading to insulin release [39].

Considering the critical role of mitochondrial oxidative metabolism in glucose-induced insulin secretion, a properly functional MCU complex is required in pancreatic β-cells for the correct functioning of this process [40]. ATP rise upon glucose stimulation is impaired when MCU is downregulated [41], resulting in a decrease in insulin secretion [40]. Interestingly, not only the pore-forming subunit MCU, but also the regulatory subunit MICU1, is important for insulin secretion. Similar to MCU silencing, MICU1 downregulation leads to a decrease in ATP levels and glucose-induced insulin secretion in pancreatic β-cells [42]. Surprisingly, the strongest reduction in insulin secretion in β-cells is observed when the regulatory subunit MICU2 is downregulated [43].

Mitochondrial dysfunction has been reported to be involved in the development and progression of insulin resistance, obesity, and diabetes [44,45]. Wright et al. investigate the MCUc component expression in obese and diabetic adipose tissue sourced from mice and humans. The authors found that in all obese and diabetic models, MCU complex components are upregulated. However, as for insulin secretion in pancreatic β-cells, not all of the MCU complex components behave similarly. However, a critical role emerges for MICU1, the only MCU complex component strongly upregulated during the transition from obesity to diabetes [46]. Finally, normal levels of MCU expression are restored in the adipose tissue of patients after bariatric surgery-induced weight loss [46].

An interesting modulation of the MCUc component expression in diabetes was observed by Belosludtsev et al. in liver cells of Sprague–Dawley rats upon streptozotocin-induced Type I diabetes. The authors found an increase in mtCa^2+^ uptake associated with a decrease in the expression of the dominant-negative subunit, MCUb [47,48].

These data suggest a key role of mitochondrial Ca^2+^ dysregulation in obesity and diabetes, highlighting the relevance of MCU as a putative therapeutic target for the treatment of these metabolic diseases.

## 5. Cancer

Dysregulation in cell proliferation and resistance against cell death are main features of cancer. Cytosolic Ca^2+^ plays a pivotal role in these phenomena. Indeed, cytosolic Ca^2+^ raises occur in concomitance with DNA synthesis, transition to G_2_/M and from anaphase to metaphase. These increases in cytosolic Ca^2+^ concentration activate transcription factors and cell-cycle relevant proteins, regulating cell proliferation. Alterations in the Ca^2+^ signaling system, involving also the mitochondrial compartment, are associated with malignant transformation [49]. 

A large number of studies have been published in the last decade associating MCU with the development and progression of different tumors. This is not surprising due to the multifarious role of mtCa^2+^ on key aspects of carcinogenesis. Oncogenes of the Bcl-2 family have been shown to reduce mtCa^2+^ loading by reducing ER Ca^2+^ levels and/or desensitizing the inositol 1,4,5 phosphate receptor (IP3R), thereby impairing apoptosis [50,51,52,53]. However, metastatic tumors have been shown to upregulate MCU, and the consequent increase of matrix Ca^2+^ favors motility and invasiveness via a ROS/HIF-1α mitochondria-to-nucleus-signaling pathway [54,55] (Figure 2). In the large number of reports highlighting a role for MCU in cancer, heterogeneity is thus the key word, not only between different cancer types but also between different cell lines of the same cancer.

### 5.1. Breast Cancer

The majority of studies about MCU and cancer are related to breast cancer. However, the picture that emerges from these studies is quite complex, with marked differences in the various cell models [56]. One of the most aggressive breast tumor subtypes is triple-negative breast cancer (TNBC). Tosatto et al. demonstrated that in TNBC, the expression of MCU correlates with tumor size and lymph node infiltration, suggesting a potential role of MCU in tumor growth and metastatic potential. Accordingly, the authors showed that in xenografts models of TNBC, MCU downregulation reduced tumor size and cell motility, as well as metastatic infiltrations. Finally, a positive correlation of MCU levels in TNBC with mROS and HIF-1α signaling was revealed [54]. Recently, a novel mechanism was proposed to explain the link between MCU and TNBC progression. Filadi et al. observed that spontaneous and sustained inositol 1,4,5-trisphosphate (IP3)-dependent Ca^2+^ oscillations occur in MDA–MB–231 cells, a widely used TNBC cell type, and they are associated with the regulation of fatty acid (FA) metabolism. Furthermore, the modulation of mitochondrial FA metabolism mediated by Ca^2+^ concentration has a deep effect on MDA–MB–231 cell migration [57]. Notably, the critical role of MCU regulation of cell metabolism in supporting tumor growth and proliferation was recently further confirmed by Fernandez Garcia et al. [58].

Curry et al. showed that MCU downregulation in MDA–MB–231 cells did not cause changes in proliferation or viability. Interestingly, distinct effects of MCU downregulation were observed on cell-death pathways. The authors showed that MCU silencing increased the cytotoxicity in MDA–MB–231 cells only in a caspase-independent way. Indeed, cell death was increased only when MCU silencing was accompanied by treatment with ionomycin, while no effects were observed in combination with the Bcl-2 inhibitor ABT–263, which is initiator of caspase-dependent cell death [59]. In this case, similar to data obtained by the same authors in other cancers, the authors postulate that impaired mtCa^2+^ uptake reduces Ca^2+^ buffering and, thus, favors Ca^2+^-dependent apoptotic pathways.

A way to induce cell death through MCU modulation was also investigated by De Mario et al. in a study that identified positive and negative MCU modulators. Among the negative modulators, benzethonium emerged as an effective compound. In MDA–MB–231 cells, benzethonium, by negatively modulating MCU, delays tumor cell growth and migration [60]. Conversely, another recent study showed that MCU upregulation can be critical for MDA–MB–231 cell survival. Xue et al. showed that MCU is required for the pro-apoptotic effect of RY10–4, a protoapigenone analog, on MDA–MB–231 cells. Treatment with this compound causes a strong MCU upregulation, which results in mtCa^2+^ overload and, finally, apoptosis [61].

Tang et al. showed that MCU overexpression is critical for breast cancer cell migration. Indeed, MCU inhibition by ruthenium red or downregulation by specific siRNA impairs MDA–MB–231 cell migration [62]. Hall et al. further confirmed that MCU overexpression was associated with poorer disease outcomes. However, they also observed an opposite effect with MICU1 since its downregulation was associated with a negative disease outcome. The authors also demonstrated that the downregulation of MCU did not affect ROS production, and that MCU was dispensable for MDA–MB–231 clonogenic cell survival [63]. Finally, Yu et al. demonstrated that MCU overexpression in MC7F, another type of human breast carcinoma line, increases the invasiveness and metastatic potential of these cells [64].

Overall, MCU plays a critical role as checkpoint of the metastatic behavior in breast cancer, thus enlightening its potential role as pharmacological target in this aggressive type of cancer.

### 5.2. Pancreatic Cancer

In pancreatic cancer, the histidine triad nucleotide binding protein 2 (HINT2) was showed to promote cell death, and Chen et al. showed that treatment with ruthenium red, an inhibitor of MCU, reduced HINT2-dependent-induced apoptosis. Interestingly, an overexpression of HINT2 increases the expression of EMRE and decreases the expression of MICU1 and MICU2. The authors hypothesized a possible mechanism in which a constitutively active channel, lacking the regulation of the MICU subunits, could account for mtCa^2+^ overload and consequent cell death [65]. In line with this study, Xie et al. showed that the overexpression of EMRE positively correlates with pancreatic ductal adenocarcinoma (PDAC) prognosis [66]. Recently, elevated mtCa^2+^ uptake was associated with metastasis formation in PDAC. The effect of MCU in PDCA metastasis is through the activation of the KEap–Nrf2 antioxidant program [67]. Overall, these studies suggest that MCU also plays a role in the pathogenesis of pancreatic cancer, in particular the structural subunit EMRE. 

### 5.3. Colon Cancer

Marchi et al. identified miRNA-25, a cancer-related MCU-targeting microRNA. Experiments performed on HeLa cells showed that miRNA-25 overexpression decreases mtCa^2+^ uptake without affecting cytosolic Ca^2+^ levels. The expression of this miRNA is upregulated in human colon cancers where MCU expression is accordingly downregulated, and this correlates with apoptotic death resistance [68]. However, recently, Yu et al. proposed that the lncRNA CERS6 antisense RNA 1 (CERS6-AS1) promotes colon cancer progression via the upregulation of MCU [69]. Recently, another miRNA was found to target MCU in colorectal cancer (CRC), miR-138-5p. In CRC, miR-138-5p is downregulated, increasing MCU expression [70]. A further study by Zeng et al. showed that mtCa^2+^ uptake promotes colorectal cancer progression through the interaction between RIPK1, a signaling molecule essential for cell survival, and MCU [71]. Although further studies are needed to clarify the role of mtCa^2+^ in colon cancer progression, these studies underly the importance of miRNA and lncRNA, instead of MCU directly, as a possible target to modulate MCU activity in pathological contests.

### 5.4. Hepatocellular Carcinoma

Ren et al. demonstrated that in HCC cells, the expression of MCU was enhanced, MICU1 was downregulated, while MICU2, MCUb, and EMRE expression levels were unaffected. Moreover, MCU upregulation is associated with poor survival and metastasis in HCC patients. The authors showed that the strong increase in mtCa^2+^ uptake promotes ROS production by downregulating NAD+, sirtuin3 (SIRT3), and superoxide dismutase 2 (SOD2) activity. High ROS levels, in turn, stimulate metalloproteinase-2 activity, increasing cell motility [72]. In a second publication by the same group, it was also shown that the regulator of the MCU complex, MCUR1, was enhanced in HCC cells. The consequent increase in mtCa^2+^ uptake leads to an increase in ROS production and ROS-dependent p53 degradation, promoting cancer cell survival [73]. Another study investigating the expression profile of long noncoding RNA in sub-lethal heat-treated HCC cells showed a downregulation of MCU. This model characterizes a transition zone of radiofrequency ablation (RFA), a treatment insufficient to kill all tumor cells, leading to residual cancer occurrence [74]. Therefore, a complex scenario is emerging regarding the mtCa^2+^ signaling in hepatocellular carcinoma, in which the role of different MCUc components must be taken in account. 

### 5.5. Other Cancer Types

A critical role of MCU in many other cancer types has emerged in recent years.

A recent work by Stejerean–Todoran et al. enlightened the role of the MCU complex in melanoma, showing that the silencing of MCU suppresses melanoma cell growth, but it promotes cell migration and invasion, reducing the sensitivity to immunotherapies [55].

High levels of MCU expression were also found in ovarian cancer, and its silencing reduces ovarian cancer cell proliferation and migration. This was correlated to a decrease in ROS production [75]. Similarly, the reduction of MCU expression in renal cell carcinoma, through the overexpression of the EF-hand domain family member D1 protein (EFHD1), and a negative regulator of MCU activity, leads to the reduction in cell migration and metastatic potential [76].

The opposite effects of MCU silencing in the above-mentioned cancers show that the modulation of mtCa^2+^ signaling can have different effects depending on the cancer-type and therefore it is not possible to formulate a general concept of how the MCU modulation affect cancer progression.

## 6. Skeletal Muscle Diseases

The molecular identification of MCU was followed by intensive studies on skeletal muscle aimed at characterizing the role of mtCa^2+^ homeostasis in this tissue, which is characterized with a specific physiology and Ca^2+^-signaling repertoire [77]. The study of the first global MCU knockout mouse model exhibited the most prominent alterations in the skeletal muscle. As expected, in this model, both resting mtCa^2+^ concentrations and stimulated mtCa^2+^ uptake were reduced. These alterations caused an impairment in mitochondrial oxidative metabolism with an increase in the phosphorylation level of pyruvate dehydrogenase, leading to a reduction in TCA cycle activity. The defective mitochondrial energetic control is responsible for the reduction in exercise performance and muscle force [29]. Mammucari et al. studied the role of MCU in adult skeletal muscle, avoiding compensatory effects that can be present in the global knockout model. MCU expression was modulated through silencing and overexpression in vivo; overexpression of MCU caused muscle hypertrophy, while the silencing of MCU led to muscle atrophy. Interestingly, the control of muscle size and trophism by MCU, which were observed in both developing and adult muscles, did not depend on the effect on aerobic metabolism, but on the regulation of two major pathways of skeletal muscle hypertrophy: IGF1-Akt and PGC-1α4 [78]. To further characterize the role of MCU in skeletal muscle physiology, Gherardi et al. generated a skeletal muscle-specific MCU knockout, characterized by the myofiber-specific impairment of mtCa^2+^ uptake. This triggered a decrease in muscle exercise performance and force, as well as a fiber-type switch, from slow to fast MHC expression. Notably, the loss of MCU rewired skeletal muscle metabolism toward fatty acid oxidation [79] (Figure 3).

Interestingly, another MCU complex component, MCUb, the dominant–negative form of MCU, plays a key role during skeletal muscle regeneration by modulating macrophage-driven stimulation and the differentiation of satellite cells after muscle damage. In particular, MCUb was shown to drive macrophage polarization from the pro-inflammatory phenotype to the anti-inflammatory phenotype, including the secretion of cytokines that promote satellite cell differentiation and fusion [80].

The role of the MCU complex in skeletal muscle physiology is critical, and mutations in the MICU1 gene were reported in human patients with a disease phenotype characterized by learning difficulties and a progressive extrapyramidal movement disorder. Clinically, the disease was characterized by early onset proximal muscle weakness, intellectual impairment, and elevated levels of serum creatine kinases. At the genetic level, different loss-of-function mutations were found, resulting in the loss of the MICU1 protein. This leads to an increased mtCa^2+^ load, increasing sensitivity to cell death stimuli but also resulting in lower cytoplasmic Ca^2+^ levels, potentially impinging on muscle contraction and synaptic transmission [81]. To further understand the mechanisms behind this neuromuscular disease, Debattisti et al. characterized patient cells and skeletal muscle-specific MICU1 knockout mice. A lack of MICU1 was associated with a low threshold for MCU-mediated Ca^2+^ uptake. Notably, MICU1 loss causes muscle atrophy and a decrease in force. The alterations in mtCa^2+^ uptake during sarcolemma injury leads to an ineffective muscle repair [82]. Recently, the neural pathogenesis was characterized by Singh et al. They generated a neuron-specific MICU1–KO mouse model showing progressive motor and cognitive degeneration. MICU1–KO neurons are more susceptible to mtCa^2+^ overload and cell death, although this is reverted by the inhibition of the mPTP [83].

MICU1 was also shown to be critical in another skeletal muscle disorder, the Barth syndrome, which is characterized by cardiolipin deficiency. Ghosh et al. utilized several Barth syndrome models, including yeast, mouse model, and patient cells. This showed that cardiolipin is required for the stability of MICU1, which is reduced in Barth syndrome patient-derived cells, together with MCU and MICU2. The reduction in mtCa^2+^ uptake results in reduced mitochondrial respiration [84].

Duchenne muscular dystrophy (DMD) is a severe neuromuscular disorder caused by mutations in the dystrophin gene, which leads to progressive degradation of skeletal muscle structure, a loss of mobility, and, eventually, premature death [85,86]. At the cellular level, the absence of dystrophin causes alterations in sarcolemma permeability, which are associated with an abnormal increase in cytCa^2+^ concentration [87]. In this context, morphological and functional alterations of mitochondria were observed. In particular, DMD has been found to be characterized by an impairment in oxidative phosphorylation, enhanced ROS production, and the opening of the mPTP, ultimately resulting in programmed cell death [88,89]. Dubinin et al. showed that changes in MCUc component expression affect mtCa^2+^ handling in this disorder. The authors observed an increased level of the dominant–negative subunit of the complex, MCUb, in skeletal muscle mitochondria of *mdx* mice, which was associated with a decrease in the rate of mtCa^2+^ uptake. Interestingly, a strong increase in the expression of adenylate translocator (ANT2), a possible component of mPTP, which was associated with a lower resistance of DMD mitochondria to mPTP induction [90], was also observed. The same group also investigated the role of mtCa^2+^ in heart mitochondria of *mdx* mice, showing, in this case, an increased rate of mtCa^2+^ uptake due to an increased MCU/MCUb ratio. Notably, an increased mtCa^2+^ efflux was also observed [91].

Finally, mtCa^2+^ uptake was found to be critical in embryonal rhabdomyosarcoma (ERMS). Indeed, MCU expression is upregulated in ERMS, and its downregulation causes mROS level decreases and an increased propensity to differentiate, inhibiting the oncogenic phenotype [92].

Overall, these data show that in skeletal muscle, the modulation of mitochondrial Ca^2+^ concentration is crucial for the control of muscle trophism and force. Interestingly, an MCU machinery geared for maximizing mitochondrial responses allows for the prompt trigger of upregulation of ATP production in cells with heavy energetic requirements during contraction. However, such potent machinery must be kept under control by efficient gatekeeping, preventing mitochondrial Ca^2+^ overload and cycling in resting conditions. For this purpose, the MICU family of regulators is crucial. Indeed, the genetic loss of MICU1 leads to severe muscle damage. Finally, the link between mitochondria and cell function is clearly not limited to metabolic control but impinges on the regulation of gene expression via the production of signaling molecules (such as ROS and metabolites). These new Ca^2+^-dependent mitochondria-to-nucleus signaling routes have been elucidated in the case of muscle trophism. They will prove crucial in a much broader set of cases linking the organelle Ca^2+^ homeostasis to the organ physiopathology.

## 7. Neurodegenerative Diseases

Neurodegenerative diseases comprise a large group of heterogeneous disorders, such as Alzheimer’s disease, Parkinson’s disease, Huntington’s disease, and amyotrophic lateral sclerosis. They are all characterized by the selective cell death of neuronal subtypes. A common feature of these disorders is mtCa^2+^ overload, which can trigger the opening of mPTP, leading to cell death [93].

### 7.1. Alzheimer’s Disease

The pathogenic mechanisms of Alzheimer’s disease (AD) are still poorly characterized, but a hallmark is the aberrant processing of the amyloid precursor protein (APP) mediated by γ-secretase. The catalytic components of γ-secretase, presenilin-1 and -2, are enriched in the mitochondria-associate ER membranes (MAMs) in cell models of AD [94]. Zampese et al. showed that the overexpression and the silencing of presenilin-2 modulate the transfer of Ca^2+^ between ER and mitochondria. In particular, the overexpression of presenilin-2 mutants found in familial AD increased the physical interaction between ER and mitochondria, increasing mtCa^2+^ entry [95]. Cheung et al. showed that familial AD presenilin-1 and -2 mutants interact with the 1,4,5-trisphosphate receptor (InsP3R) to promote the Ca^2+^ release from ER [96]. Importantly, this leads to mtCa^2+^ overload and an increase in the open probability of mPTP [97]. Notably, the pathogenesis of familial AD was ameliorated in the mouse model with the suppression of InsP3R-mediated Ca^2+^ signaling [98]. Interestingly, amyloid-beta and prion peptides can also induce the release of Ca^2+^ from the ER [99]. The reduction of NCLX activity, which is another way to increase the susceptibility to Ca^2+^-induced cell death, was found in AD neurons, supporting the notion of mtCa^2+^ overload being crucial in AD progression [100].

### 7.2. Parkinson’s Disease

Parkinson’s disease (PD) is a neurological disorder associated with the loss of dopaminergic neurons in the substantia nigra, characterized by typical motor symptoms, including tremors and muscle rigidity [101]. Models of PD caused by alpha–synuclein overexpression reveal an excessive mtCa^2+^ uptake, enhancing ROS production and impairing oxidative metabolism [102]. Mutations in PINK1 are also causative of PD, and Gandhi et al. investigated the alterations in mtCa^2+^ homeostasis in PINK1-deficient neurons. PINK1 deficiency causes mtCa^2+^ overload, increasing ROS production and impairing respiration, resulting in neuronal cell death [103]. The genetic and pharmacological inactivation of MCU in a pink1-mutant zebrafish prevents dopaminergic neuronal cell death via the rescue of mitochondrial respiration [104]. Interestingly, mutations in the E3 ubiquitin-ligase Parkin also cause familial PD. Matteucci et al. showed that Parkin is required for the proteasome-mediated degradation of MICU1. Probably in an indirect way, MICU2 stability was also affected upon Parkin overexpression. This study suggests that Parkin loss could contribute to the impairment in mtCa^2+^ handling through its regulation of MICU proteins [105]. Finally, mutations in leucine-rich repeat kinase 2 (LRRK2), a common genetic cause of PD, increased the expression of MCU and MICU2, enhancing mtCa^2+^ uptake in cortical neurons and familial PD cells [106].

### 7.3. Huntington’s Disease

Huntington’s disease (HD) is a progressive neurodegenerative disorder in which the huntingtin gene is expanded by CAG triplet repeat, leading to an N-terminal polyglutamine strand of variable length [107]. Panov et al. showed a reduction in mtCa^2+^ uptake in HD cells from patients combined with a reduction in membrane potential [108]. However, the susceptibility to the mPTP opening is increased in mitochondria isolated from HD cells [109]. Another study conducted on striatal neurons of HD showed that mitochondria were unable to handle large Ca^2+^ loads, possibly due to the increasing sensitivity to mPTP, which reduces membrane potential, leading to Ca^2+^ release [110].

### 7.4. Amyotrophic Lateral Sclerosis

Amyotrophic lateral sclerosis (ALS) is a progressive and fatal motoneuron disease characterized by muscle weakness, atrophy, and, eventually, paralysis that leads to death [111]. The role of mtCa^2+^ in ALS changes during the progression of the disease. Indeed, in cultured embryonic motor neurons from the ALS mouse model, MCU is upregulated, and its pharmacological inhibition is protective against excitotoxicity. Instead, MCU expression is reduced in motor neurons from the symptomatic ALS mouse model [112]. However, another study showed that in motor neurons of the ALS end stage, MCU and MICU1 were upregulated [113]. Thus, a different MCU modulation can be protective depending on the distinct phase of the disease.

## 8. Conclusions and Future Perspectives

In summary, since the molecular identification of MCU, research on this essential signaling component of mitochondria has dramatically increased, with several hundreds of papers providing a deeper insight into the association with other proteins, the regulation of its activity, and its role in the physiology of a broad variety of tissues. It has become clear that the essential constituent of the pore region, MCU itself, is part of a larger complex, and that the molecular composition of the latter provides wide flexibility to the molecular machinery of a specific cell type, which is in agreement with the physiological properties of the cell. It is also becoming increasingly clear that alterations of this finely tuned process have effects on events as diverse as metabolism, cell death, and inflammation. In this review, we have summarized the increasing evidence associating dysregulation of MCU with the pathogenesis of diseases of high prevalence and social impact. While mechanistic insight in most cases still needs to be fully elucidated, these data highlight the mitochondrial calcium-signaling machinery as a promising pharmacological target and suggest that the clarification of the role of the individual molecular components may lead to new drugs of high precision in these diseases.

## Figures and Tables

**Figure 1 biomolecules-13-01304-f001:**
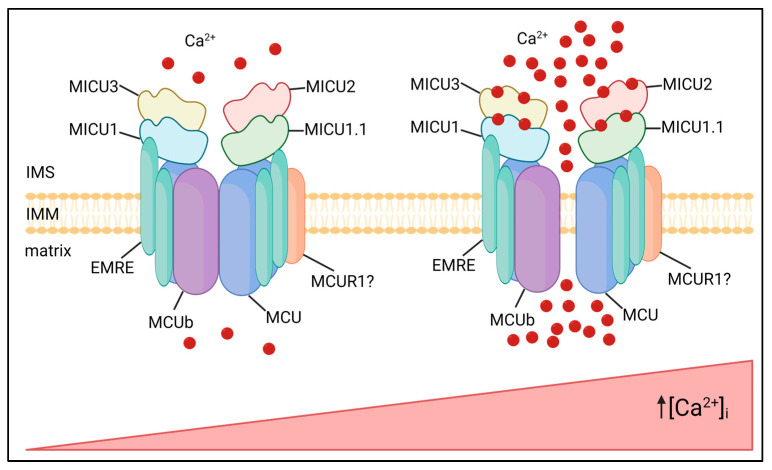
MCU complex composition and activity. The MCU complex localizes in the IMM and is composed of pore-forming subunits MCU and MCUb, regulatory subunits, MICU family, and the structural protein EMRE. The MICU proteins sense increases in [Ca^2+^]_i_ via EF-hand domains, undergoing conformational changes and allowing Ca^2+^ entry into the mitochondrial matrix. In this representative image of the MCU complex, all the known isoforms of MICUs are shown, even if their expression is cell-type specific.

**Figure 2 biomolecules-13-01304-f002:**
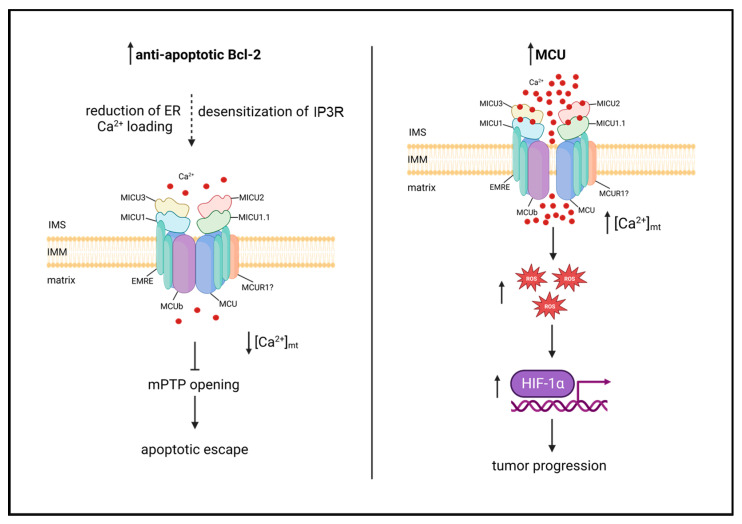
MtCa^2+^ dysfunctions in cancer. (Left panel): Anti-apoptotic Bcl-2 proteins reduce mtCa^2+^ loading by reducing ER Ca^2+^ loading and/or desensitizing IP3R, impairing apoptosis. (Right panel): MCU is upregulated in metastatic tumors, and the consequent increase in mtCa^2+^ uptake promotes tumor progression via ROS/HIF-1α mitochondria-to-nucleus signaling pathway.

**Figure 3 biomolecules-13-01304-f003:**
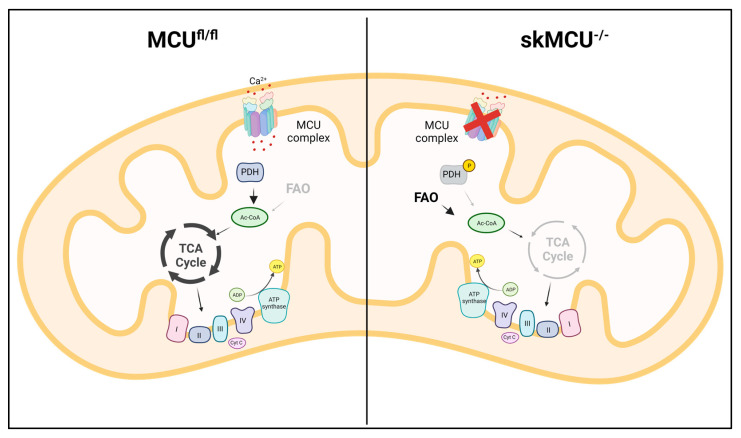
MCU deletion in skeletal muscle rewires metabolism towards FA utilization. In MCU^-/-^ muscles, the loss of mtCa^2+^ uptake leads to PDH inactivity, impairment in pyruvate oxidation, and decrease in TCA cycle activity. The impairment in glucose oxidation is partially compensated by an increase in FA oxidation.

## Data Availability

Not applicable.

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
