# Peer review of "The Mitochondrial Calcium Uniporter (MCU): Molecular Identity and Role in Human Diseases"

_biomolecules, 2023, doi:10.3390/biom13091304_

Round 1

Reviewer 1 Report

This review article provides a new perspective on the structural characteristics of MCUs and their relationship to various diseases. It is expected to be a useful review for researchers engaged in basic and clinical calcium research. However, the following revisions and additions are necessary for acceptance.

L23 1. MCU complex structure and function

In this section, the differences between cardiac muscle, skeletal muscle, and brain are mentioned, but other tissues (smooth muscle/organs) are not mentioned. This review also mentions diseases involving organs other than heart, skeletal muscle, and brain. Therefore, the description should be organized by tissue and cell type.

L30  Figure 1.

Is it possible for the left and right MICU isoforms to be different in the same tissue as shown in the figure?

L53 MCU/MCUb ratio

L56 MCUb/MCU ratio ã€€ Consistency is necessary.

L85

What part of "negative regulation" is referred to here?

What does "see below" mean?

L93-95 

The description of references 13, 14, and 15 is insufficient. The relationship between each of the findings is unclear.

L161-164 

Reference should be provided. Also, the description of obesity should be more detailed.

L183 Fig.2  left side I3PR → IP3R

L206  Is MDA-MB-231 used in the references #44 ?

L215 

Calcium ions are not superscripted.

L226

Are TNBC cells and MDA-MB-231 cells different?

L226-235 

This paragraph should be more descriptive and organized. Furthermore, interestingly, surprisingly, etc. are repeated.

L266  HCC

Pre-abbreviated description required.

L290 

What is the difference between the positive effects (increased MCU hypertrophy, etc.) and the negative effects (increased mito uptake in MICU1KO  atrophy/lower tension, increased MCU in disease models, etc.) on skeletal muscle due to differences in MCU or mitoCa levels?

I would like to know your views, not just a list of them.

Author Response

Reviewer 1

This review article provides a new perspective on the structural characteristics of MCUs and their relationship to various diseases. It is expected to be a useful review for researchers engaged in basic and clinical calcium research.

Thank you.

However, the following revisions and additions are necessary for acceptance.

L23 1. MCU complex structure and function

In this section, the differences between cardiac muscle, skeletal muscle, and brain are mentioned, but other tissues (smooth muscle/organs) are not mentioned. This review also mentions diseases involving organs other than heart, skeletal muscle, and brain. Therefore, the description should be organized by tissue and cell type.

Since the different ratio of the MCU complex components are not elucidated in all tissues and cell type, we just mentioned the tissues that have been deeply investigated. However, we added a statement explaining this and the reference to other studies that can be useful for readers interested in this aspect.

L30  Figure 1.

Is it possible for the left and right MICU isoforms to be different in the same tissue as shown in the figure?

We added a statement in the figure legend saying that this is a representative image and that the MICUs expression depend on the cell-type.

L53 MCU/MCUb ratio

L56 MCUb/MCU ratio ã€€ Consistency is necessary.

Done

L85

What part of "negative regulation" is referred to here?

What does "see below" mean?

The reviewer is right indicating that this statement is not clear, so we changed it.

L93-95 

The description of references 13, 14, and 15 is insufficient. The relationship between each of the findings is unclear.

We changed this paragraph adding more details.

L161-164 

Reference should be provided. Also, the description of obesity should be more detailed.

Done

L183 Fig.2  left side I3PR → IP3R

Done

L206  Is MDA-MB-231 used in the references #44 ?

Yes, it is. We now clarified this in the text.

L215 

Calcium ions are not superscripted.

Done

L226

Are TNBC cells and MDA-MB-231 cells different?

MDA-MB-231 cells are a widely used type of TNBC cells. We now clarified this aspect in the text.

L226-235 

This paragraph should be more descriptive and organized. Furthermore, interestingly, surprisingly, etc. are repeated.

Done

L266  HCC

Pre-abbreviated description required.

Done

L290 

What is the difference between the positive effects (increased MCU→ hypertrophy, etc.) and the negative effects (increased mito uptake in MICU1KO → atrophy/lower tension, increased MCU in disease models, etc.) on skeletal muscle due to differences in MCU or mitoCa levels?

I would like to know your views, not just a list of them.

We added a short paragraph in the end of this part with our view.

Reviewer 2 Report

The review article by D´Angelo and Rizzuto starts with a short summary of the major structural features of the mitochondrial calcium uniporter (MCU), discovered in 2011 by the lab of the authors as well as by another group. This was a major breakthrough in the field as it turned out that MCU consisting of a complex assembly of subunits is involved in important signaling reactions controlling a wide number of physiological processes. This is reflected by the large number of publications following the initial identification of the MCU.

The major part of the review covers the available knowledge of the role of the calcium uniporter in a number of human diseases. It is clear from the available data that failure of the uniporter can lead to very different effects in different tissues and even within the same tissue or cell type depending on which MCU subunit is affected and the special state of the cell or tissue.  So, it is not possible to come up with a general concept how malfunction of MCU contributes to disease development. However, gathering information using modern technologies including tissue and cell type specific gene editing will certainly help to develop specific tools to correct for the malfunction of MCU.

This article is well written, structured and contains clear graphical illustrations and will certainly be very well received by the scientific community.

Minor comments:

In a number of cases paragraphs end with the statement as shown below.

line 107

The precise role of the .......still remains to be fully elucidated

e.g. Line 140

However, further studies are needed.....

line 158

the MCU complex influence ......still needs to be fully elucidated

line 202

although further studies will be needed

line 278

Further studies are needed

line 400

further studies are needed

This is redundant and not needed as it is clear that to get the complete story more research has to be done.

Grammar:

line 266

shoud be past tense - is should be replaced with was

line 281

An emerging role is emerging ?

line 325,326

learning differences, mentioned twice

line 77,79

on the other hand should be left out and start with Low ..... as the next sentence starts with on the other hand, the same also in line173,176

Too long sentence

e.g. 416-423

Author Response

Reviewer 2

The review article by D´Angelo and Rizzuto starts with a short summary of the major structural features of the mitochondrial calcium uniporter (MCU), discovered in 2011 by the lab of the authors as well as by another group. This was a major breakthrough in the field as it turned out that MCU consisting of a complex assembly of subunits is involved in important signaling reactions controlling a wide number of physiological processes. This is reflected by the large number of publications following the initial identification of the MCU.

The major part of the review covers the available knowledge of the role of the calcium uniporter in a number of human diseases. It is clear from the available data that failure of the uniporter can lead to very different effects in different tissues and even within the same tissue or cell type depending on which MCU subunit is affected and the special state of the cell or tissue.  So, it is not possible to come up with a general concept how malfunction of MCU contributes to disease development. However, gathering information using modern technologies including tissue and cell type specific gene editing will certainly help to develop specific tools to correct for the malfunction of MCU.

This article is well written, structured and contains clear graphical illustrations and will certainly be very well received by the scientific community.

Thank you.

Minor comments:

In a number of cases paragraphs end with the statement as shown below.

line 107

The precise role of the .......still remains to be fully elucidated

e.g. Line 140

However, further studies are needed.....

line 158

the MCU complex influence ......still needs to be fully elucidated

line 202

although further studies will be needed

line 278

Further studies are needed

line 400

further studies are needed

This is redundant and not needed as it is clear that to get the complete story more research has to be done.

 Many of these redundant statement have been removed.

Grammar:

line 266

shoud be past tense - is should be replaced with was

 Done

line 281

An emerging role is emerging ?

Changed.

line 325,326

learning differences, mentioned twice

Removed.

line 77,79

on the other hand should be left out and start with Low ..... as the next sentence starts with on the other hand, the same also in line173,176

 Done.

Too long sentence

e.g. 416-423

Changed.

Reviewer 3 Report

This is a reasonably well written review from one of the leading groups in the field. The authors summarized the known data on the structure of the mitochondrial calcium uniporter and its role in the development of human pathologies. I have a couple of comments and suggestions for its content.

1. Authors need to add an introduction section that describes the problem and goal that the authors set for themselves in this review. This is important, as it will allow the reader to be immersed in the topic, to show its importance. In this case, it is also important to highlight the originality of the work, since there have been a lot of reviews on the presented topic lately.

2. In the past few years, a lot of data has been accumulated on tissue-specific rearrangements in the calcium uniporter system (both at the level of proteins and coding genes) in diabetes, neuromuscular diseases (such as Duchenne dystrophy) and other pathologies, which is accompanied by changes in the kinetics of calcium transport and affects the functioning of mitochondria, as well as the course of pathology in general. This will complement the job well.

Author Response

Reviewer 3

This is a reasonably well written review from one of the leading groups in the field. The authors summarized the known data on the structure of the mitochondrial calcium uniporter and its role in the development of human pathologies.

Thank you.

 I have a couple of comments and suggestions for its content.

  1. Authors need to add an introduction section that describes the problem and goal that the authors set for themselves in this review. This is important, as it will allow the reader to be immersed in the topic, to show its importance. In this case, it is also important to highlight the originality of the work, since there have been a lot of reviews on the presented topic lately.
  2. In the past few years, a lot of data has been accumulated on tissue-specific rearrangements in the calcium uniporter system (both at the level of proteins and coding genes) in diabetes, neuromuscular diseases (such as Duchenne dystrophy) and other pathologies, which is accompanied by changes in the kinetics of calcium transport and affects the functioning of mitochondria, as well as the course of pathology in general. This will complement the job well.

We added an introduction paragraph with the characteristic suggested by the reviewer, enlightening also the tissue-specific changes of the MCU reflected in changes in the kinetics, suggested in the second point.  We also discussed more in details the suggested diseases in point 2 (L 211-214; L391-408)

Round 2

Reviewer 3 Report

The authors adequately responded to my comments and improved the manuscript.

Author Response

Thank you.